# On the Application and Impact of $\epsilon$-DP and Fairness in Ambulance Engagement Time Prediction

**Selene Cerna & Catuscia Palamidessi**
Inria and École Polytechnique (IPP), Palaiseau, France
{selene-leya.cerna-nahuis,catuscia.palamidessi}@inria.fr

## Abstract

This study elaborates on a complete pipeline for the development of a private and fair Machine Learning (ML) model to predict ambulance engagement time. It was shown that sensitive variables reduced their impact on model building with Random Forest as the differential privacy budget ($\epsilon$) decreased with the GRR and Geometric mechanisms. Also, the application of the Reweighing fairness mechanism negatively affected fairness in private models. Finally, it is possible to keep firefighters' and victims' privacy, recovering an ML model with good performance.

## 1 Introduction and Related Work

In recent years, Artificial Intelligence (AI) has accelerated the development of various social sectors (health, education, environment, etc). However, in areas like civil security, there is still much to be developed. For instance, many Fire Departments (FDs) face high operational loads and resource constraints (Guyeux, 2022), i.e., as the population grows, so does the number of incidents (cardiac arrests, suicides, traffic accidents, etc.). If there aren't enough resources like ambulances, it can lead to system breakdowns and result in human, economic, and material losses. Fortunately, some FDs have long been recording their interventions, creating valuable data sources to exploit with AI. Taking advantage of these sources, our study contributes with *(i) the development of an intelligent model for predicting Ambulance Engagement Time (AET)*, which allows identifying if the AET will be higher than the average time of all ambulances in the last 3 hours. Thus, it will be possible to recognize trends and seasonality in ambulance departures, dynamically reorganize them between nearby centers, avoid breakdowns, and keep people protected. To the author's knowledge, there are works on the optimization, some operations forecasting, and a gap analysis on firefighter information technologies literature (Arcolezi et al., 2022b; Cerna et al., 2021; Morello et al., 2020; Agrawal et al., 2020; Lian et al., 2019; Weidinger, 2022). Nevertheless, none of them developed a predictive model for AET. Also, our study includes *(ii) an impact analysis of Differential Privacy (DP) (Dwork, 2006) and Reweighing (Rew) (Calders et al., 2009) mechanisms on model performance*. DP is applied to preserve victims' and firefighters' privacy, but it can introduce or increase an existing bias (Fioretto et al., 2022; Pujol et al., 2020). To tackle this, we will apply Rew and analyze two fairness metrics, which is an unexplored process in this area. The background is given in A.1.

## 2 Methodology and Results

The dataset has 106520 samples (ambulance departures) and 33 variables. A detailed data description is given in A.2. From here, we highlight that the target (*iTime*) is a binary indicator, where class 1 represents an AET greater than the average of the last 3 hours, and class 0 otherwise. Also, 4 sensitive discrete variables are identified: intervention type (*typeIntv*), number of professional (*nbP*) and volunteer (*nbV*) firefighters, and professionalism indicator (*iPro*), which is the protected variable. The unprivileged group (iPro=0) means nbV>nbP and the privileged group (iPro=1) otherwise.

For contributions *(i)* and *(ii)*, we will develop the following experiments: **E1)** Recover the best base model with RandomForestClassifier and the best configuration with Bayesian optimization, which will be reused in the next models. **E2)** Correct the bias according to iPro with Rew. This is because iPro's groups could naturally generate a bias in the model, i.e., there are more volunteers with basic training and few with full training which could result in more and less AET, respectively.

Table 1: Summary of results of fairness and performance measures obtained for the 4 experiments.

| Experiment | Model | Privacy budget ($\epsilon$) | AC | AC iPro=0 | AC iPro=1 | F1 | DI | SEDF |
|---|---|---|---|---|---|---|---|---|
| E1 | Base | - | 0.78 | 0.77 | 0.78 | 0.77 | 3.36 | 1.21 |
| E2 | Base + Rew. | - | 0.75 | 0.78 | 0.71 | 0.75 | 1.63 | 0.49 |
| E3 | GRR | 9 | 0.76 | 0.75 | 0.77 | 0.76 | 3.64 | 1.29 |
| | GRR | 6 | 0.76 | 0.76 | 0.77 | 0.76 | 3.61 | 1.28 |
| | GRR | 0.5 | 0.76 | 0.76 | 0.77 | 0.76 | 3.57 | 1.27 |
| | GRR + Geometric | 9 | 0.76 | 0.76 | 0.77 | 0.76 | 3.67 | 1.30 |
| | GRR + Geometric | 6 | 0.73 | 0.71 | 0.77 | 0.72 | 2.92 | 1.07 |
| | GRR + Geometric | 0.5 | 0.76 | 0.76 | 0.77 | 0.76 | 3.51 | 1.26 |
| E4 | GRR + Rew | 9 | 0.76 | 0.74 | 0.78 | 0.76 | 5.28 | 1.66 |
| | GRR + Rew | 6 | 0.76 | 0.75 | 0.78 | 0.76 | 4.60 | 1.52 |
| | GRR + Rew | 0.5 | 0.76 | 0.76 | 0.77 | 0.76 | 3.55 | 1.27 |
| | GRR + Geometric + Rew | 9 | 0.75 | 0.75 | 0.76 | 0.75 | 3.60 | 1.28 |
| | GRR + Geometric + Rew | 6 | 0.76 | 0.75 | 0.77 | 0.76 | 4.26 | 1.45 |
| | GRR + Geometric + Rew | 0.5 | 0.76 | 0.76 | 0.77 | 0.76 | 3.54 | 1.26 |

**E3)** Assess the impact of privacy level on models ($\forall \epsilon \in L = \{0.1, 0.2, ...1, 2, ..., 15\}$). To protect victims' privacy, we will sanitize typeIntv with GRR (Kairouz et al., 2016). To protect firefighters' privacy, we will test 2 approaches: when nbP and nbV are sanitized with Geometric (Ghosh et al., 2008), since both are discrete variables; and sanitized with GRR, since both can be considered multi-categorical. **E4)** Assess the impact of privacy level on models with Rew to correct the bias. Finally, since DP algorithms are randomized, we will report average results over 20 runs on private models. For variables typeIntv, nbP, and nbV, $\epsilon$ will be divided uniformly. To measure model performance, we will use Accuracy (AC) and F1-Score (F1). To estimate model fairness, we will use Disparate Impact (DI) and Smoothed Empirical Differential Fairness (SEDF).

A detailed description of settings and results is given in A.3. Table 1 summarizes our findings. In general, private models (with and without Rew) presented higher AC than the fair base model but lower than the base. If we analyze AC per iPro's group, combinations of GRR and Geometric showed less stable performance than GRR only. Also, as $\epsilon$ decreased (high privacy), nbP and nbV reduced their importance, leading to a stabilization of the models' performance. Fig. 1 shows the reduction of sensitive variables' impact for 3 values of $\epsilon$, considering GRR only.

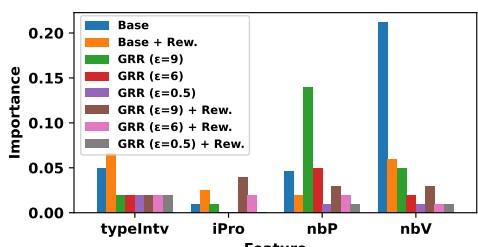

Figure 1: Sensitive variables' impact.

Last, results showed that private models without Rew were fairer than when Rew is added. Moreover, applying DP during preprocessing enhanced model fairness as $\epsilon$ decreased. This contrasts with Bagdasaryan et al. (2019); Ganev et al. (2022), in which DP was applied during the inprocessing stage and its impact was negative. Besides, we searched and used the best base model hyperparameters for all experiments, but in real life, an optimal hyperparameter search should be performed to recover the best configuration and deploy it to production (de Oliveira et al., 2023).

## 3 CONCLUSION

This study elaborated a complete pipeline for the development of a private and fair ML model to predict AET. We conclude that it is possible to recover an ML model with good AC and fairness, protecting firefighters' and victims' privacy by applying DP in the preprocessing stage. Our findings were: a) GRR alone degrades model performance less than when combined with Geometric. b) Rew application negatively affects private models' fairness, but not the base model. c) In high privacy regimes, DP enhanced models' fairness. And, d) The stabilization of models' AC is due to the loss of typeIntv, nbP, nbV, and iPro importance during the training. The experimentation source codes are available at PredictingAET. However, the data are private and not open source. Finally, some future directions are: analyzing feature importance in private models and assessing the DP impact, using different encoding for sensitive variables (label, one hot encoding, etc), and determining whether the same results are obtained using different types of classifiers (linear, boosting, neural network, etc).

ACKNOWLEDGEMENTS

This work was supported by the European Research Council (ERC) project HYPATIA under the European Union's Horizon 2020 research and innovation program. Grant agreement n. 835294. The authors would also like to thank Service Départemental d'Incendie et de Secours du Doubs (SDIS 25) for their great collaboration and continuous feedback.

URM STATEMENT

The authors acknowledge that at least one key author of this work meets the URM criteria of ICLR 2023 Tiny Papers Track.

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

# A APPENDIX

## A.1 BACKGROUND

In this section, basic notions related to ML, DP and Fairness techniques, that we use, are given.

**Supervised Learning**. A dataset $D$ contains pairs $(x_i, y_i)$ of input and true label, respectively, and a function $g$ outputs a predicted label $g(x_i)$. The training process of $g$ is done to capture patterns from $x_i$ while minimizing the loss function $\mathcal{L}(g(x_i), y_i)$ until obtaining the best model $g$. An example of technique is **Random Forest**, an ensemble learning method of type bagging (Breiman, 2001; Pedregosa et al., 2011).

**Differential Privacy**. A randomized mechanism $M$ provides $(\epsilon, \delta)$-DP for every set of outputs $S$, and for any neighbouring datasets $D$ and $D'$ differing in one element, if and only if $M$ satisfies $Pr[M(D) \in S] \leq e^{\epsilon} Pr[M(D') \in S] + \delta$. The privacy budget is represented by $\epsilon \geq 0$, the smaller the more privacy. If $\delta = 0$, $M$ gives $\epsilon$-DP in strict definition, and $\delta > 0$ gives freedom to violate $\epsilon$-DP for some low probability events. For multi-step mechanisms $M = \{M_1, M_2, ..., M_m\}$, 3 properties must be considered: *a) Post-processing*, where any additional processing to $M$'s output or the composition $M_1(M_2(.))$ still satisfies $\epsilon$-DP; *b) Sequential composition*, where mechanisms $M$, sequentially performed on a dataset with their respective $\epsilon_i$-DP, will provide $(\sum_{i=1}^{m} \epsilon_i)$-DP; and, *c) Parallel composition*, where mechanisms $M$, applied on a disjointed subset of the entire dataset and with their respective $\epsilon_i$-DP, will provide $(max\{\epsilon_1, ..., \epsilon_m\})$-DP (Dwork, 2006; Zhu et al., 2017).

**Geometric**. It is a DP mechanism, defined as $M(D') = f(D') + z$, where $f(D')$ is the true query result and $z$ is a random variable from the double geometric distribution $\frac{1 - e^{-\epsilon \Delta^{-1}}}{1 + e^{-\epsilon \Delta^{-1}}} e^{-\epsilon |z| \Delta^{-1}}$. For counting functions, the sensitivity ($\Delta$) is 1, which will be our case (Ghosh et al., 2008; Bellamy et al., 2018).

**Direct Encoding**. Also known as Generalized Randomized Response (GRR), that extends the Randomized Response DP mechanism for cases where $k \geq 2$, being $k$ the size of a set of categories $A = \{v_1, ..., v_k\}$ of a given attribute. Thus, GRR(v) outputs the true value with probability $p$, and any other value $v' \in A$ such that $v' \neq v$ with probability $1 - p$. More precisely, the perturbation is defined as a conditional function: if $y = v$, then $p = \frac{e^{\epsilon}}{e^{\epsilon} + k - 1}$; or if $y \neq v$, then $q = \frac{1}{e^{\epsilon} + k - 1}$, $\forall y \in A$ (Kairouz et al., 2016; Arcolezi et al., 2022a).

**Reweighing**. It is a mechanism that considers independence constraint to build a fair classifier. Let's say a sample $x$ has a binary protected attribute $B$ with value $b$ ($b \in dom(B) = \{0, 1\}$). If its expected probability ($P_{exp}$) is higher than its actual probability ($P_{act}$), the bias will be towards the negative class (class $-$) for $B = b$ and the weights will be assigned to $b$ with respect to positive class (class $+$): $W(B = b|x(Class) = +) = \frac{P_{exp}(b \wedge +)}{P_{act}(b \wedge +)}$. The weight of $b$ for class $-$ will be: $W(B = b|x(Class) = -) = \frac{P_{exp}(b \wedge -)}{P_{act}(b \wedge -)}$. The same calculation is applied for weights of $\bar{b}$ for classes $+$ and $-$ (Calders et al., 2009; Kamiran & Calders, 2011; Bellamy et al., 2018).

**Disparate Impact**. It is a fairness metric, which measures the positive proportions in the predicted outcomes. Values less than and greater than 1 indicate positive and negative bias, respectively. The ideal value should be close to 1 (Nielsen, 2020).

**Smoothed Empirical Differential Fairness**. It is a fairness metric, which compares the differential of smoothed probability between groups and provides a closely related privacy guarantee to differential privacy. The ideal value should be close to 0 (Foulds et al., 2020). In our experiments, we will use the method published in Bellamy et al. (2018) with its default parameters.

## A.2 Data Collection and Analysis

The main source is a list of ambulance departures from 01/01/2018 to 12/31/2021, coming from the 71 centers of the Departmental Fire and Rescue of Doubs (SDIS 25), in France. The terms *ambulances*, *firefighters*, and *victims* represent the various types of engines in the organization, agents with various types of training, and people affected in the incidents, respectively. Since our goal is to predict AET, we filtered 14 internal input variables (center, distance, engine, experience, nbP, nbV, iPro, typeIntv, day, hour, isNight, month, weekday, and year) and the target variable, which is the ambulance's engagement time in minutes. AET is the calculated time from when firefighters receive the alert to respond to an intervention until the moment they return to their center. In addition, we extracted 15 variables in total from external sources such as traffic indicators from Bison-Futé and meteorological variables from 3 stations (Bâle, Dijon, and Nancy) of Météo-France.

Next, we cleaned incomplete data and labeled categorical variables, recovering 106520 samples in total. Besides, to improve the quality of our data, for each sample (at time $t$), we generated 3 input variables (iAvg1h, iAvg2h, and iAvg3h) to indicate if the average engagement time in the last 3 hours ($t-1$, $t-2$, and $t-3$) was greater than the average of all departures made in the last month.

Table 2: Description of variables.

| Type | Variable | Description | Values |
|------|----------|-------------|--------|
| Input - Internal | center | Fire station from which the ambulance departed. | {1,2,...,71} |
| | distance | Distance between the fire station and the incident scene (km). | [0.001-7103] |
| | engine | Type of vehicle (ambulance) used. | {1,2,3,4,5} |
| | experience | Number of firefighter departures in the last month. | {0,1,...,108} |
| | nbP | Number of professional firefighters. | {1,2,3,4} |
| | nbV | Number of volunteer firefighters. | {1,2,3,4,5} |
| | iPro | Crew professionalism indicator, where: iPro=1 means nbV≤nbP, iPro=0 means nbV>nbP. | {0,1} |
| | typeIntv | Type of intervention (altered consciousness, cardio-respiratory arrest, imminent delivery, etc). | {0,1,...,36} |
| | day | Day of the intervention. | {1,2,...,31} |
| | hour | Hour of the intervention. | {0,1,...,23} |
| | iAvg1h | Indicator for the average engagement time, where: iAvg1h=1 means that the average engagement time of all ambulance departures in the last hour is greater than the average of all those made in the last month, iAvg1h=0 means otherwise. | {0,1} |
| | iAvg2h | Indicator for the average engagement time, where: iAvg1h=1 means that the average engagement time of all ambulance departures in the last 2 hours is greater than the average of all those made in the last month, iAvg1h=0 means otherwise. | {0,1} |
| | iAvg3h | Indicator for the average engagement time, where: iAvg1h=1 means that the average engagement time of all ambulance departures in the last 3 hours is greater than the average of all those made in the last month, iAvg1h=0 means otherwise. | {0,1} |
| | isNight | Indicator to identify if it is night or day. | {0,1} |
| | month | Month of the intervention. | {1,2,...,12} |
| | weekday | Weekday of the intervention. | {0,1,...,6} |
| | year | Year of the intervention. | {2018, 2019, 2020, 2021} |
| Input - Traffic (Bison-Futé) | traficDepart | Traffic level indicator on exit roads. | {0,1,2,3} |
| | traficReturn | Traffic level indicator on arrival roads. | {0,1,2,3} |
| Input - Weather (Météo-France) | cloudinessBale | Cloudiness registered by a station in Bâle. | [0-101] |
| | humiBale | Humidity registered by a station in Bâle. | [18-100] |
| | humiDijon | Humidity registered by a station in Dijon. | [1-100] |
| | humiNancy | Humidity registered by a station in Nancy. | [14-100] |
| | precipBale | Precipitation registered by a station in Bâle. | [-0.1-22.1] |
| | precipDijon | Precipitation registered by a station in Dijon. | [-0.1-24.3] |
| | precipNancy | Precipitation registered by a station in Nancy. | [-0.1-8.8] |
| | tempBale | Temperature registered by a station in Bâle. | [257.05-311.45] |
| | tempDijon | Temperature registered by a station in Dijon. | [262.55-312.05] |
| | tempNancy | Temperature registered by a station in Nancy. | [261.15-312.55] |
| | windBale | Wind speed registered by a station in Bâle. | [0.0-15.1] |
| | windDijon | Wind speed registered by a station in Dijon. | [0.0-16.6] |
| | windNancy | Wind speed registered by a station in Nancy. | [0.0-16.9] |
| Output - Target | iTime | Indicator for ambulance engagement time, where: $iTime = 1$ means that the ambulance engagement time is greater than the average engagement time of all ambulances in the last 3 hours, $iTime = 0$ means otherwise | {0,1} |

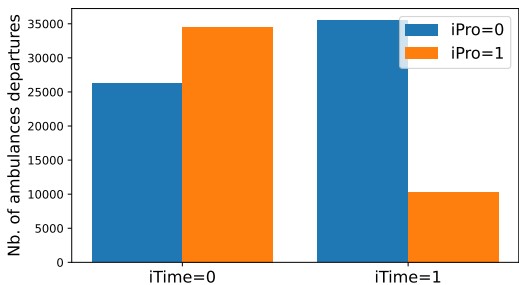

Figure 2: Number of samples per class (iTime) and group (iPro).

This allows us to establish trends over the years. Additionally, to define our classes, our target was converted into a binary indicator for classification (iTime), where class 1 represents an engagement time greater than the average of the last 3 hours, and class 0 otherwise. The conversion of the continuous target variable to a binary target variable is due to the fact that we want a response as an indicator with respect to a previous time (3 hours) and not an approximate prediction in minutes. In this way, we can recognize the seasonality of ambulances' engagement time throughout the day, since more interventions are performed during the day than at night, and therefore, ambulances' engagement time varies. Thus, our final dataset, called DS-AET, consists of 33 variables described in detail in Table 2.

Finally, when analyzing the dataset more deeply, we detected 4 sensitive variables: the type of intervention (typeIntv), sensitive for victims because it would reveal what problem they had (cardiac arrest, traffic accident, suicide, etc.); the number of professional (nbP) and volunteer (nbV) firefighters who departed with the ambulance, sensitive for the center and for firefighters because if the intervention was not successful and there was only one firefighter in the ambulance, they would be easily identified based on intervention's datetime; and the crew professionalism indicator (iPro), which is the protected variable. In iPro, the unprivileged group (iPro=0) means that there were more volunteers than professionals in the ambulance (nbV>nbP), and the privileged group (iPro=1) otherwise. Although iPro depends on nbV and nbP, maintaining its existence as an indicator will allow us to analyze fairness metrics and use Rew as a bias mitigation mechanism. Also, the difference between professionals and volunteers is that the former dedicate their time exclusively to the fire department and the latter partially, which in practice gives an advantage in experience and training to professionals. The latter is reflected in Fig. 2, which shows the distribution of samples per class and iPro's group. Here, we see that there is an imbalance in both the classes and iPro groups, more exactly, the majority of ambulances had more volunteers (iPro=0) and the ambulances that spent more time engaged (iTime=1) were those that had more volunteers (iPro=0) than professionals (iPro=1). This could generate a bias in the predictions, since during the learning process the classifier would pay more attention to the majority group, i.e., it could generate a prejudice towards volunteers, believing that if there are more volunteers than professionals in an ambulance, the predicted engagement time would be longer. However, in real life, there are cases where ambulances have experienced volunteers and novice professionals, which would reject the previous prediction. This analysis is important to highlight since we must protect victims' and firefighters' privacy and avoid the generation or increment of bias towards certain groups in our intelligent models.

## A.3  COMPLEMENTARY EXPERIMENTS AND RESULTS

The present study aims to develop an intelligent model to predict AET (iTime), using the Random-ForestClassifier technique. This technique is well suited for tabular data and allows us to extract the feature importance easily. In addition, given that there are 4 sensitive variables of victims and firefighters (nbV, nbP, typeIntv, and iPro) in DS-AET, we will apply 2 differential privacy mechanisms (GRR and Geometric) to protect individuals' privacy and 1 fairness mechanism (Rew) to correct the bias towards the unprivileged group. Next, we will assess the impact of these mechanisms on the performance and fairness of the built models and analyze the behavior of the sensitive variables with the importance retrieved from the classifier. Finally, we will recover the best model.

In general, for model learning and evaluation, samples from DS-AET are divided into 2 groups: before 30/06/2021 (training set) and after 01/07/2021 (testing set). To describe model performance, the AC and F1 metrics will be used. To measure model fairness, the DI and SEDF metrics will be computed. We will calculate AC since it is the most common metric for binary classification and allows us to recognize how many correct predictions we got; F1 because our dataset is imbalanced and we want to obtain a good balance between precision and recall; DI because we want to measure group fairness, more precisely, whether there is any indirect discrimination against volunteers or professionals when the ambulance spends a long time engaged; and SEDF since it provides an intersectional definition of fairness for groups and with a similar interpretation to DP.

In the following, we detail the 4 experiments that will be conducted:

- E1) We will build a base model to predict AET. For this, we divide DS-AET into training and testing sets. We fit our model with the training set and run a search for recovering the best hyperparameter configuration with Bayesian optimization (Bergstra et al., 2013). Table 3 describes the defined search space for RandomForestClassifier, the number of iterations performed, and the loss function applied to predictions based on the testing set. It also shows the best configuration recovered, which will be used in all other models of the following experiments.

- E2) We will apply Rew to the base model. For this, we divide DS-AET into training and testing sets. We fit Rew with iPro and iTime from the training set to calculate sample weights. Later, these weights will be used during model training, considering the best hyperparameter configuration recovered in E1. In this way, Rew allows us to correct the bias generated by iPro and analyze the fairness of our model according to the metrics computed with the testing set.

- E3) We will develop 2 groups of private models with GRR and Geometric. In the first group, typeIntv will be sanitized with GRR since it is categorical and nbP and nbV will be sanitized with Geometric since these are count variables. In the second group, typeIntv, nbV, and nbP will be sanitized with GRR only, since nbV and nbP can be considered categorical too with defined space. For both groups, we will develop an iterative process to test 24 privacy budgets ($\forall \epsilon \in L = \{0.1, 0.2, ...1, 2, ..., 15\}$). For each privacy budget, we will run 20 different seeds due to the randomness of DP algorithms. The processes performed during an iteration with a budget and a seed will be: split DS-AET into training and testing sets; preprocess typeIntv, nbV, and nbP from the training set with the budget divided uniformly for the three variables; recalculate iPro based on the new values of nbV and nbP; built a model with the preprocessed data and with the best hyperparameter configuration recovered in E1; extract the importance of features according to the model; and, calculate performance and fairness metrics on the testing set. To obtain the final results for each private budget, we will average its resulting metrics and the importance of its features over the 20 runs.

- E4) We will develop the same 2 groups of private models in E3, with the difference that we will add Rew immediately after preprocessing variables typeIntv, nbV, and nbP, and recalculating iPro. This way, we seek to correct a potential bias increase due to the application of GRR and Geometric mechanisms (Bagdasaryan et al., 2019; Ganev et al., 2022).

The results are shown in Fig. 3 and Fig. 4, which show the performance (AC) and fairness (DI and SEDF), respectively, of all the models created in the 4 experiments, highlighting the difference between private models based on the combination of GRR and Geometric (left) and GRR alone (right). Also, in Table 4, we detail the resulting metrics per experiment, model, and privacy level, rounded to 2 decimal places. For experiments E3 and E4, 8 values of $\epsilon$ are exemplified. And in Fig. 5, which illustrates the importance of the features most used by the classifiers for the 4 experiments, exemplifying 3 levels of $\epsilon$ in private models, and zooming on the 4 sensitive variables. From here, we deduce:

Table 3: Definition of the search space for RandomForestClassifier, the best-recovered hyperparameter configuration, the defined loss function, and the number of iterations for Bayesian Optimization.

| Search space | Best configuration | | |
|---|---|---|---|
| n_estimators: [200-600] | 200 | | |
| max_depth: [10-15] | 14 | | |
| max_features: [0.5-1] | 0.8 | | |
| max_samples: [0.5-1] | 0.95 | | |
| class_weight: {0:1-5, 1:1-5} | {0:1, 1:2} | | |
| **Loss function** | $\mathcal{L} = \begin{cases} \text{(AC iTime=0) * (AC iTime=1)} & \text{, if AC iTime=0} > 0.7 \text{ and AC iTime=1} > 0.7 \\ \text{(AC iTime=0) + (AC iTime=1)} & \text{, otherwise} \end{cases}$ | | |
| **Number of iterations** | 50 | | |

Table 4: Performance and fairness results per experiment (Exp.) and with 8 privacy levels. AC is the model accuracy, AC iPro=0 is the privileged group accuracy, AC iPro=1 is the unprivileged group accuracy, and |AC iPro=1 - AC iPro=0| is the absolute value of the difference in group accuracies.

| Exp. | Model | Privacy ($\epsilon$) | Metrics | | | | | | |
|---|---|---|---|---|---|---|---|---|---|
| | | | AC | AC iPro=0 | AC iPro=1 | F1 | DI | SEDF | \|AC iPro=1 - AC iPro=0\| |
| E1 | Base | - | 0.78 | 0.77 | 0.78 | 0.77 | 3.36 | 1.21 | 0.01 |
| E2 | Base + Rew. | - | 0.75 | 0.78 | 0.71 | 0.75 | 1.63 | 0.49 | 0.07 |
| E3 | GRR | 15 | 0.77 | 0.77 | 0.77 | 0.77 | 3.45 | 1.24 | 0.00 |
| E3 | GRR | 12 | 0.76 | 0.76 | 0.77 | 0.76 | 3.68 | 1.30 | 0.01 |
| E3 | GRR | 9 | 0.76 | 0.75 | 0.77 | 0.76 | 3.64 | 1.29 | 0.02 |
| E3 | GRR | 6 | 0.76 | 0.76 | 0.77 | 0.76 | 3.61 | 1.28 | 0.01 |
| E3 | GRR | 3 | 0.76 | 0.76 | 0.77 | 0.76 | 3.55 | 1.27 | 0.01 |
| E3 | GRR | 1 | 0.76 | 0.76 | 0.77 | 0.76 | 3.55 | 1.27 | 0.01 |
| E3 | GRR | 0.5 | 0.76 | 0.76 | 0.77 | 0.76 | 3.57 | 1.27 | 0.01 |
| E3 | GRR | 0.1 | 0.76 | 0.76 | 0.77 | 0.76 | 3.54 | 1.26 | 0.01 |
| E3 | GRR + Geometric | 15 | 0.77 | 0.77 | 0.77 | 0.77 | 3.35 | 1.21 | 0.00 |
| E3 | GRR + Geometric | 12 | 0.77 | 0.76 | 0.77 | 0.76 | 3.51 | 1.25 | 0.01 |
| E3 | GRR + Geometric | 9 | 0.76 | 0.76 | 0.77 | 0.76 | 3.67 | 1.30 | 0.01 |
| E3 | GRR + Geometric | 6 | 0.73 | 0.71 | 0.77 | 0.72 | 2.92 | 1.07 | 0.06 |
| E3 | GRR + Geometric | 3 | 0.76 | 0.75 | 0.77 | 0.76 | 3.90 | 1.36 | 0.02 |
| E3 | GRR + Geometric | 1 | 0.76 | 0.76 | 0.77 | 0.76 | 3.54 | 1.26 | 0.01 |
| E3 | GRR + Geometric | 0.5 | 0.76 | 0.76 | 0.77 | 0.76 | 3.51 | 1.26 | 0.01 |
| E3 | GRR + Geometric | 0.1 | 0.76 | 0.76 | 0.77 | 0.76 | 3.53 | 1.26 | 0.01 |
| E4 | GRR + Rew | 15 | 0.77 | 0.76 | 0.78 | 0.77 | 4.53 | 1.51 | 0.02 |
| E4 | GRR + Rew | 12 | 0.77 | 0.75 | 0.78 | 0.76 | 4.63 | 1.53 | 0.03 |
| E4 | GRR + Rew | 9 | 0.76 | 0.74 | 0.78 | 0.76 | 5.28 | 1.66 | 0.04 |
| E4 | GRR + Rew | 6 | 0.76 | 0.75 | 0.78 | 0.76 | 4.60 | 1.52 | 0.03 |
| E4 | GRR + Rew | 3 | 0.76 | 0.76 | 0.77 | 0.76 | 3.50 | 1.25 | 0.01 |
| E4 | GRR + Rew | 1 | 0.76 | 0.76 | 0.77 | 0.76 | 3.52 | 1.26 | 0.01 |
| E4 | GRR + Rew | 0.5 | 0.76 | 0.76 | 0.77 | 0.76 | 3.55 | 1.27 | 0.01 |
| E4 | GRR + Rew | 0.1 | 0.76 | 0.76 | 0.77 | 0.76 | 3.55 | 1.27 | 0.01 |
| E4 | GRR + Geometric + Rew | 15 | 0.77 | 0.76 | 0.77 | 0.76 | 3.45 | 1.24 | 0.01 |
| E4 | GRR + Geometric + Rew | 12 | 0.76 | 0.76 | 0.77 | 0.76 | 3.68 | 1.30 | 0.01 |
| E4 | GRR + Geometric + Rew | 9 | 0.75 | 0.75 | 0.76 | 0.75 | 3.60 | 1.28 | 0.01 |
| E4 | GRR + Geometric + Rew | 6 | 0.76 | 0.75 | 0.77 | 0.76 | 4.26 | 1.45 | 0.02 |
| E4 | GRR + Geometric + Rew | 3 | 0.76 | 0.75 | 0.78 | 0.76 | 5.25 | 1.65 | 0.03 |
| E4 | GRR + Geometric + Rew | 1 | 0.76 | 0.76 | 0.77 | 0.76 | 4.16 | 1.42 | 0.01 |
| E4 | GRR + Geometric + Rew | 0.5 | 0.76 | 0.76 | 0.77 | 0.76 | 3.54 | 1.26 | 0.01 |
| E4 | GRR + Geometric + Rew | 0.1 | 0.76 | 0.76 | 0.77 | 0.76 | 3.53 | 1.26 | 0.01 |

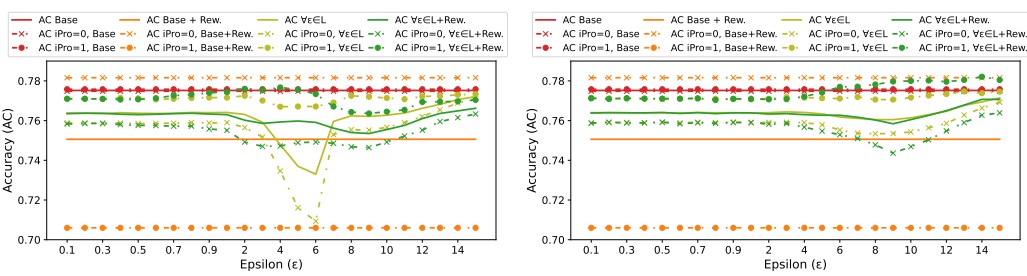

(a) Performance of GRR and Geometric-based models    (b) Performance of GRR-based models

Figure 3: Model accuracy (AC) and accuracy per group (AC iPro) of the base model (red), the base model with Rew (orange), private models (olive green), and private models with Rew (dark green). On the left, models with GRR and Geometric. On the right, models with GRR.

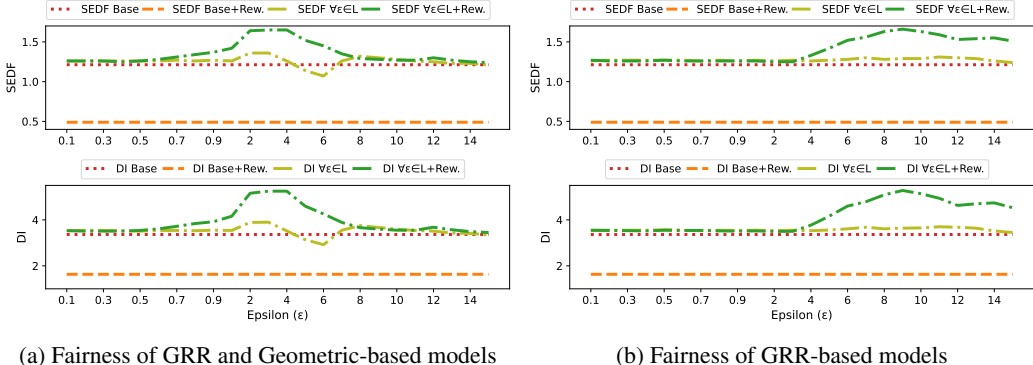

(a) Fairness of GRR and Geometric-based models     (b) Fairness of GRR-based models

Figure 4: Fairness measures comparison (DI and SEDF) of the base model (red), the base model with Rew (orange), private models (olive green), and private models with Rew (dark green). On the left, models with GRR and Geometric. On the right, models with GRR.

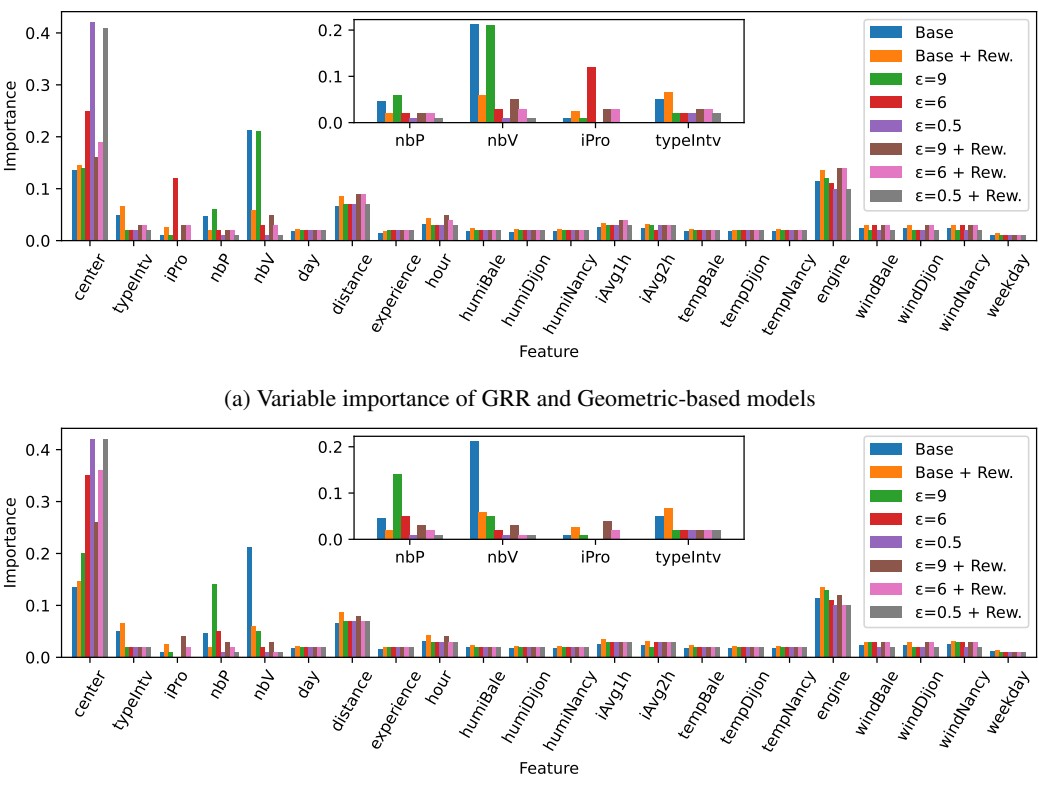

(a) Variable importance of GRR and Geometric-based models

(b) Variable importance of GRR-based models

Figure 5: The 22 most important variables and their behavior when adding DP and Rew, zooming on nbP, nbV, iPro, and typeIntv. It is shown the base model (blue), the base model with Rew (orange), 3 private models with $\epsilon = [9, 6, 0.5]$ (green, red, and purple), and 3 private models with $\epsilon = [9, 6, 0.5]$ and Rew (brown, pink, and gray). At the top, models with GRR and Geometric. At the bottom, models with GRR.

First, in Table 4, we observe that our base model is biased and iPro groups have a balanced accuracy. The metrics of the base model with Rew show that fairness was improved but accuracy deteriorated, especially the accuracy of the privileged group (AC iPro=1). In private models, as $\epsilon$ decreased the total accuracy of the models decreased until it stabilized. However, the difference between the accuracies of groups did not increase, i.e., in high privacy regimes, there was no continuous negative

impact of DP on group performance. In Fig. 3, we see that the accuracy of private models is lower than the base model but higher than the base model with Rew. Also, the accuracy of private models with GRR was degraded less and it was more stable than private models with GRR and Geometric.

Second, in Fig. 4, we notice that, in general, private models' fairness was far from that of the base model with Rew. However, private models' fairness without Rew was higher than with Rew and close to that of the base model when $\epsilon$ decreased, i.e., in high privacy regimes, DP enhanced models' fairness and the impact of Rew was negative on private models. In addition, private models with GRR reach stable fairness metrics more quickly than models with GRR and Geometric as the privacy level decreased.

Third, from Fig. 5, we see that the stabilization of the performance and fairness in private models was due to the importance reduction of variables nbV, nbP, iPro, and typeIntv as the level of privacy increased. Besides, the importance of nbV was negatively and rapidly affected when we used GRR and not Geometric, and other variables such as center increased in importance as $\epsilon$ decreased. This is why, private models with GRR quickly obtained better fairness metrics.

Finally, the works of Bagdasaryan et al. (2019); Ganev et al. (2022) studied the impact of DP on privileged and unprivileged groups by applying it during the inprocessing stage and suggest that DP generates a disparate impact on the accuracy of the groups as privacy level increases. However, in our study, defining explicit performance and fairness metrics, we demonstrate that if we apply DP during the data preprocessing, DP will improve model fairness for high privacy regimes, and the difference in accuracies between the groups will stabilize. Moreover, it is recommended not to apply a bias correction mechanism like Rew after applying DP in preprocessing, as it will reduce the fairness of the model. Additionally, while we searched and considered the hyperparameter configuration of the base model for all other models, in real life, it is advisable to perform a hyperparameter search for each new model, as discussed in de Oliveira et al. (2023). Although there will be higher resource consumption, the best configuration would be retrieved to be deployed in production.

