# OpenReview forum: "On the application and impact of ε-DP and fairness in ambulance engagement time prediction"
_ICLR.cc/2023/TinyPapers — Submitted to Tiny Papers @ ICLR 2023_

### Official Review · Reviewer_gcCK · 2023-03-25

**Confidence:** 3

**Summary Of Contributions:**

This study elaborates on a complete pipeline for the development of a private and fair machine learning model to predict ambulance engagement time.

**Rating:**

Great Start (GS): a submission which meets some of the reviewing criteria but has room for improvement

**Strengths And Weaknesses:**

In general I think this paper is quite meaningful in real-world reliable machine learning by considering fairness and DP-privacy. The analysis sounds quite interesting.

However, this reviewer feels like current submission lacking clarity. Further the paper organization could be improved. Several suggestions are in *suggested changes*.

**Suggested Changes:**

I would suggest author

- fixing several typos such as
>  This results in long response times and leaves a locality unprotected.
- revising experimental section in Sec 2.
- zooming Fig1 and Fig2, since they are too small to read.

---

> ### Author Response · Authors · 2023-05-30
> **Reply to Reviewer gcCK**
>
> Thank you very much for your suggestions, they helped us to improve our article. We have corrected the typos, revised and elaborated more experiments, and improved the clarity of the images.

---

### Official Review · Reviewer_ZHz2 · 2023-03-30

**Confidence:** 3

**Summary Of Contributions:**

Using the random classifier model, the authors attempt to predict the ambulance engagement time while ensuring fairness and privacy.

**Rating:**

Great Start (GS): a submission which meets some of the reviewing criteria but has room for improvement

**Strengths And Weaknesses:**

The authors attempt to solve an interesting problem.

However, it's hard to understand the authors' contributions as the write-up is not cohesive or concise. The sentences are hard to connect and each raises lots of questions.
The authors don't define the terms victim and agent but frequently use them in the write-up.

Additionally, the authors don't explain their modeling choices, which makes it hard to appreciate and fairly assess their contributions. For example, why is the number of professional versus volunteer firefighters considered a sensitive attribute? Why do authors only consider firefighters and not EMTs and their demographic attributes?
If nbV and nbP determine iProf, why are they part of the variables used in predicting ambulance engagement time?
Among other questions, why is the engagement time binarized?

Lastly, the figures critical to assessing the results are not visible. It's hard to make out the writing/captions, therefore, hard to appreciate the authors' contributions.


**Suggested Changes:**

The authors should provide more support for their modeling choices and experimental setups.
They should also make the write-up flow clearer and more cohesive to ensure better reading and comprehension of the key ideas presented.

---

> ### Author Response · Authors · 2023-05-30
> **Reply to Reviewer ZHz2**
>
> Thank you very much for your questions and recommendations. We have added a more detailed explanation for the processes that we performed and improved the quality of the images.

---

### Comment · Area_Chair_yyCR · 2023-06-04
**Revised version**

This work meets the threshold for archival, contains the URM statement and is deanonymized

---

### Meta-Review · Area_Chair_yyCR · 2023-04-08

**Recommendation:** Invite to revise
**Confidence:** 5

**Metareview:**

Predict the ambulance engagement using RF while ensuring fairness and privacy.

An interesting and vital area of research, limited by the clarity and organisation of the paper. Although the analysis sounds interesting, the figures are extremely hard to read. It is hard to identify correctness and reproducibility without a clear explanation and discussion of choices.


**Summary:**

Predict the ambulance engagement using RF while ensuring fairness and privacy.

**Comments And Feedback To The Authors:**

Please read the reviewers' comments. Maybe get someone to proof read your paper. This paper can be published provided the required work is done.

**Reason For Not Giving A Higher Recommendation:**

Unfortunately, the writing of the paper is not clear, and the results are not very visible. So it's hard to identify correctness and reproducibility.

**Reason For Not Giving A Lower Recommendation:**

N/A

---

> ### Author Response · Authors · 2023-05-30
> **Reply to Area Chair yyCR**
>
> Thank you very much for allowing a re-submission of our manuscript, with an opportunity to address the reviewers' comments. We have updated our manuscript according to the reviewers' recommendations.

---

### Decision · Program_Chairs · 2023-04-10

Revision accepted; invite to archive